  

EMBO
Molecular Medicine

# Gut microbiota interact with breast cancer therapeutics to modulate efficacy

Alana A Arnone[1], Katherine Ansley[2,3], Arielle L Heeke[3,4], Marissa Howard-McNatt[3,5] & Katherine L Cook [1,3,5]✉

## Abstract

**The gut microbiome, or the community of microorganisms residing in the gastrointestinal tract, has emerged as an important factor in breast cancer etiology and treatment. Specifically, the impact of gut bacterial populations on breast cancer therapeutic outcomes is an emerging area of research. The microbiota's role in modifying the pharmacokinetics of chemotherapy and endocrine-targeting therapies can alter drug efficacy and toxicity profiles. In addition, the gut microbiome's capacity to regulate systemic inflammation and immune responses may influence the effectiveness of both conventional and immunotherapeutic strategies for the treatment of breast cancer. Overall, while the bidirectional interactions between the gut microbiome and breast cancer therapies are still being studied, its impact is increasingly recognized. Future research may provide more definitive insights and help develop personalized therapeutic strategies to harness the microbiome to improve breast cancer treatment outcomes.**

**Keywords** Microbiome; Endocrine-targeting Therapies; Chemotherapy; Immune Checkpoint Blockade; Cyclin-dependent Kinase 4/6 Inhibitors
**Subject Categories** Cancer; Microbiology, Virology & Host Pathogen Interaction

## Introduction

Breast cancer is a complex disease that affects millions of women worldwide. In 2020, there were ~2.3 million new cases of breast cancer globally and 685,000 deaths, making breast cancer the most frequently diagnosed cancer and the leading cause of cancer death among women (Sedeta et al, 2023). By 2040, the number of new breast cancer cases is projected to increase to around 2,964,197, representing a 31% increase from the 2,260,127 cases reported in 2020 (Arnold et al, 2022). This rise in incidence is attributed to factors such as population growth, aging, and the adoption of lifestyle behaviors that increase cancer risk (Sedeta et al, 2023). The primary risk factors for breast cancer include older age, high body mass index (BMI) or obesity, tobacco exposure, physical inactivity, high-fat diet, early menarche, late age at first full-term pregnancy, shorter breastfeeding duration, use of hormonal menopausal therapy, breast density, and family history of breast cancer (Britt et al, 2020; Lei et al, 2021). Recent studies indicate that the pathogenesis and prognosis of breast cancer is not only dictated by the properties of the malignant cells but also by immune and microbial parameters (Terrisse et al, 2023). Thus, the gut microbiome has emerged as a potential mediator for breast cancer, with growing evidence suggesting that the composition and function of gut bacteria may influence breast cancer development and progression, as well as the response to treatment.

The gut microbiome comprises 10–100 trillion symbiotic microbial cells, including bacteria, phages, viruses, fungi, and protozoa (Cresci et al, 2019; Marchesi and Ravel, 2015). The majority of the healthy gut microbiome is composed of anaerobic bacteria from two major phyla, Firmicutes and Bacteroidetes (Arumugam et al, 2011; Cresci et al, 2019; Sekirov et al, 2010; Zoetendal et al, 2008). The Bacteroidetes and Firmicutes phyla are highly conserved in virtually all individuals, although the relative proportions of these phyla can vary (Lozupone et al, 2012). However, at the species level, which encompasses ~35,000 bacterial species, interindividual microbial composition differs greatly from the phylum level (Bull and Plummer, 2014). This diversity in microbiota plays an essential role in host physiology, including nutrient and mineral absorption, production of high-energy metabolites, such as short-chain fatty acids (SCFA), immune homeostasis, and protection against pathogenic bacteria (Francescone et al, 2014). The gut microbiota maintains a symbiotic relationship with the gut mucosa, resulting in substantial metabolic, immunological, and gut protective functions under normal conditions. This mutualistic interaction contributes to various aspects of host health, including nutrient and drug metabolism, immunomodulation, and maintaining a protective barrier within the gastrointestinal tract. The balance in this symbiotic relationship is crucial for overall gut homeostasis and the well-being of the host (Jandhyala et al, 2015).

Dysbiosis, or microbial imbalance, can elevate breast cancer risk by disrupting estrogen metabolism and reducing the production of anticancer metabolites (Mikó et al, 2019; Ruo et al, 2021). This dysbiosis can lead to chronic inflammation, DNA damage, and activation of oncogenic pathways, allowing bacteria like

[1]Department of Cancer Biology, Wake Forest University School of Medicine, Winston-Salem, NC, USA. [2]Department of Internal Medicine, Section on Hematology and Oncology, Wake Forest University School of Medicine, Winston-Salem, NC, USA. [3]Atrium Health Wake Forest Baptist Comprehensive Cancer Center, Winston-Salem, NC, USA. [4]Department of Solid Tumor Oncology and Investigational Therapeutics, Levine Cancer Institute, Atrium Health, Charlotte, NC, USA. [5]Department of Surgery, Wake Forest University School of Medicine, Winston-Salem, NC, USA. ✉E-mail: klcook@wakehealth.edu

**Glossary**

| | | | |
|---|---|---|---|
| Alpha (α) diversity | a term referring to species diversity within a sample. Different alpha-diversity calculations include the Chao1, Shannon, and Simpson diversity indices. These measurements account for the number of different species present (richness) and the relative abundances of each species (evenness) within a sample. | | microbial diversity, an overgrowth of pathogenic microbes, or a decline in beneficial microbial functions. |
| | | Metagenomic sequencing | the genomic and functional analysis of microbial communities (bacterial, fungal, viral, protozoa) by direct analysis of extracted DNA. |
| Beta (β) diversity | or dissimilarity index, a term to describe the degree of difference in species composition between communities or samples. | 16S rRNA sequencing | a method used to identify and classify Bacteria and Archaea in environmental or biological samples by sequencing specific regions of the 16S ribosomal RNA gene, which is present in all prokaryotes and contains both highly conserved and variable regions. |
| Dysbiosis | A term used to describe an imbalance or maladaptation of the microbiome, often associated with a disease state. Dysbiosis can involve a loss of | | |

*Fusobacterium nucleatum* and certain strains of *Escherichia coli* to promote carcinogenesis (Doocey et al, 2022; Li, 2023). Furthermore, intratumoral bacteria can affect cancer phenotypes, such as enhancing the metastatic ability of malignant cells, thereby contributing to disease advancement (Zhao et al, 2023). Dysbiosis-associated bacteria produce metabolites like lipopolysaccharide (LPS) and colibactin, which drive cancer progression through inflammatory and mutagenic pathways. LPS, produced by Gram-negative bacteria like *E. coli*, promotes tumor growth (Soto-Pantoja et al, 2021), while colibactin from *E. coli* can cause DNA double-strand breaks, leading to mutations (Sun et al, 2023). Dysbiosis also depletes beneficial bacteria that produce SCFAs, such as *Faecalibacterium prausnitzii* and *Roseburia* species, thereby reducing SCFAs' anti-inflammatory and anticancer effects (Shrode et al, 2023). The depletion of commensal bacteria, such as SCFA-producing species (Álvarez-Mercado et al, 2023), not only weakens the body's natural defense mechanisms by compromising immune system homeostasis but also creates an environment where pathogenic bacteria may have a greater influence on cancer progression (Doocey et al, 2022; Lu and Tong, 2024).

Preclinical studies have shown *Bacteroidetes fragilis* toxin can increase the aggressiveness of breast cancers, induce self-renewal in breast cancer cells, and initiate metastatic dissemination to distant organs (Parida et al, 2021). In addition, *Fusobacterium nucleatum* has also been shown to promote breast tumor growth and metastatic progression, possibly by colonizing breast tumors and by preventing the accumulation of tumor-infiltrating T cells in the tumor microenvironment (Parhi et al, 2020). Furthermore, alterations in gut microbial diversity have been observed in women without breast cancer versus those with breast cancer. A clinical study showed that fecal microbiota in postmenopausal breast cancer patients exhibited reduced α-diversity and an overall different β-diversity compared to controls (Goedert et al, 2015). Specifically, *Clostridia*, *Enterobacterium*, *Lactobacillus*, *Bacteroides*, and *Escherichia coli* were found in higher abundance in breast cancer patients compared to "healthy" controls (Goedert et al, 2015). It is important to note in these studies the characteristics of the control population. In this case, it was women with a normal screening mammogram result. *Akkermansia muciniphila* correlates with smaller tumors, and more than half of breast cancer patients lack *A. muciniphila* in their gut (Nandi et al, 2023). Comparison of the gut microbial composition of women with benign breast lesions, breast cancer, and control group revealed increased levels of *Porphyromonas* and *Peptoniphilus* genera in women with breast

cancer, while *Escherichia* and *Lactobacillus* were enriched in women with benign breast lesions, compared to the control (Ma et al, 2022). Furthermore, a lower gut microbial diversity is observed in breast cancer patients compared to control participants, with breast cancer patients displaying an enrichment in Firmicutes and a depletion in Bacteroidetes phyla (Liu et al, 2023a). Breast cancer patients also exhibited a decreased enrichment of *Odoribacter sp.*, *Butyricimonas sp.*, and *Coprococcus sp.* in contrast to the healthy controls (Bobin-Dubigeon et al, 2021). These findings underscore the complex and multifaceted role of the gut microbiome in breast cancer development and progression, highlighting its potential as a target for preventive and therapeutic strategies.

The intricate relationship between the gut microbiome and breast cancer risk becomes even more evident when considering how gut dysbiosis contributes to obesity and related diseases, both of which are established risk factors for breast cancer development (Avtanski et al, 2023). Multiple studies have shown significant differences between the gut microbiota of healthy women compared to women with breast cancer, with some showing an overlap with obese microbiota (Parida and Sharma, 2019). Further, the level of adiposity is positively associated with reduced microbial diversity and a shift in the abundance of dominant species (Davis, 2016). This shift in gut microbes increases potential pathobionts like *Alistipes finegoldii* and decreases beneficial bacteria like *Bacteroides vulgatus* and *Akkermansia muciniphila* in obese individuals (Kang et al, 2023). Additionally, dysbiosis modifies adipokines; increasing leptin and decreasing adiponectin, which are linked to higher and lower breast cancer risk, respectively (Nehme et al, 2022; Yu et al, 2019). Obesity-induced gut dysbiosis also contributes to carcinogenesis through increased chronic systemic inflammation, as obese individuals have impaired intestinal barrier integrity and increased pro-inflammatory markers (Gaber et al, 2024). Obesity-associated dysbiosis is associated with altered estrogen metabolism as well, leading to increased circulating estrogen levels, potentially increasing breast cancer risk (Argolo et al, 2018; Gaber et al, 2024). Therefore, the pathological link between obesity and breast cancer risk and prognosis is tied to metabolic, inflammatory, and hormonal dysregulation, suggesting a link between the gut microbiome and obesity-mediated breast cancer risk.

Recent studies suggest the relationship between the gut microbiome and breast cancer risk may also play a crucial role in modulating the efficacy and side effects of cancer therapies (Di Modica et al, 2021; Schettini et al, 2023; Zhou et al, 2022). By influencing drug metabolism, the gut microbiota can impact both

therapeutic outcomes and toxicity profiles (Nandi et al, 2023). Specific bacteria like *Ruminococcaceae* (family) and *Faecalibacterium* (genus), are associated with enhanced systemic and antitumor immune responses, potentially improving the efficacy of immunotherapies (Kang et al, 2024). In addition, certain gut bacteria may improve chemotherapy response by strengthening immune priming and specific antitumor immunity (Knisely et al, 2023). Given the role of gut microbiota in estrogen metabolism, it may also influence the effectiveness of hormone treatments for hormone-sensitive breast cancers. Therefore, the composition of the gut microbiome may be predictive of therapy response and quality of life in breast cancer patients. Understanding the interplay between the gut microbiome and breast cancer treatments opens avenues for personalized microbiome-based therapeutic approaches to improve clinical management.

# Chemotherapy

Chemotherapy, often used in combination with surgery, radiation, and other targeted therapies, is designed to target and kill rapidly dividing cells by interfering with various stages of the cell cycle, thereby preventing cancer cells from growing and dividing (Bilenduke et al, 2022). The timing of systemic chemotherapy for operable breast cancer includes adjuvant therapy administered after surgery and neoadjuvant therapy given before surgery (Shien and Iwata, 2020). Standard chemotherapy includes either an anthracycline (e.g., doxorubicin (DOX)), taxanes (paclitaxel, docetaxel)), cyclophosphamide, and/or antimetabolites such as capecitabine, or platinum-based chemotherapies (Moo et al, 2018).

Preclinical and clinical studies demonstrate that chemotherapy profoundly impacts the gut microbiome, reducing microbial diversity, and disrupting gut homeostasis. This disruption compromises gut homeostasis, which can alter cancer treatments' effectiveness and toxicity. Such imbalance also increases intestinal permeability and weakens immune responses, making patients more susceptible to infections and inflammation (Fakharian et al, 2023; Oh et al, 2021; Roggiani et al, 2023). Triple-negative breast cancer (TNBC) patients undergoing neoadjuvant chemotherapy display elevated plasma LPS-binding protein concentrations than breast cancer patients untreated at the time of tumor resection, supporting the impact of chemotherapy on gut permeability (Bawaneh et al, 2022). Chemotherapy depletes beneficial bacteria like *Bifidobacterium* and *Lactobacillus*, while promoting the growth of opportunistic pathogens such as *Clostridium difficile* (Ciernikova et al, 2021). Further, breast cancer patients undergoing chemotherapy also experience a decreased relative abundance of mucin-degrading bacteria like *Akkermansia* compared to healthy controls (Bilenduke et al, 2022). These disruptions are linked to gastrointestinal mucositis, a common and painful side effect of chemotherapy, characterized by inflammation and ulceration of the digestive tract. Further, the gut microbiome's composition can also affect chemotherapy outcomes, including treatment efficacy (Oh et al, 2021). Certain microbial compositions are associated with better responses to chemotherapy, while others are linked to increased toxicity and side effects. For example, in two breast cancer cohorts, authors observed that several bacterial species (i.e., *Clostridia* such as *C. hathewayi, C. clostridioforme, C. symbosium, C. aldenense, C. citroniae, E. ramosum*, as well as *Veillonella* species,

and *Eisenbergiella massiliensis* or *E. tayi*) associated with antibiotic uptake or resistance to immune checkpoint blockade (ICB) therapy became less abundant after adjuvant or neoadjuvant treatment with anthracyclines and taxanes (Terrisse et al, 2021). ICB associations with gut microbiome will be discussed in a separate section below. In regards to gut microbiome associations with adverse events, a study in lung cancer patients found that the relative abundance of *Bacteroides nordii*, and *Ruminococcus sp_5_1_39BFAA* were associated with severe adverse events following chemotherapy (Zhao et al, 2021). However, the impact of chemotherapy on gut microbiota and its association with the development of adverse events in breast cancer patients remains under-explored (Bruce et al, 2021). Understanding the interactions between cancer therapy and the gut microbiome provides opportunities for potential interventions, such as probiotic supplementation, dietary modifications, or even fecal microbiota transplantation (FMT), to mitigate chemotherapy-related toxicities and improve treatment outcomes (Knisely et al, 2023).

## Cyclophosphamide

Cyclophosphamide, an alkylating agent that cross-links DNA, preventing cell division, is commonly used in combination chemotherapy regimens for breast cancer treatment, both as adjuvant therapy and for advanced disease (Ogino and Tadi, 2024). Several findings highlight the complex interplay between cyclophosphamide, the gut microbiome, and the host immune system in cancer treatment. Cyclophosphamide alters the gut microbial composition, decreasing Bacteroidetes while increasing Firmicutes phyla proportional abundance in tumor-bearing mice (Xu and Zhang, 2015). By damaging the gut mucosa and increasing intestinal permeability, cyclophosphamide allows bacteria like *Lactobacilli* and *Enterococci* to translocate to secondary lymphoid organs, such as the spleen (Viaud et al, 2013). These translocated bacteria, particularly *Enterococcus hirae* and *Barnesiella intestinihominis*, stimulate specific immune responses, including the generation of pathogenic T helper 17 (pTH17) cells and memory Th1 responses. This immune stimulation enhances cyclophosphamide's efficacy by increasing cytotoxic T-cell accumulation in tumors (Daillère et al, 2016; Viaud et al, 2013). The drug's effectiveness is significantly reduced in germ-free mice or those treated with antibiotics to eliminate Gram-positive bacteria, highlighting the crucial role of the microbiome. Notably, specific Gram-positive bacteria, including *Lactobacillus johnsonii, Lactobacillus rhamnosus*, and *Enterococcus hirae*, increase in abundance after cyclophosphamide treatment. Gut colonization with *Eubacterium rectale, Eubacterium eligens, Akkermansia muciniphila, Bifidobacterium longum, Collinsella aerofaciens*, and *Alistipes shahii* favors cyclophosphamide efficacy in mice (Terrisse et al, 2021). These findings underscore the importance of the gut microbiome in mediating cyclophosphamide's anticancer effects and suggest potential strategies for enhancing treatment efficacy through microbiome modulation.

## Anthracyclines and/or taxanes

Anthracyclines are primarily used in the treatment of TNBC, and certain high-risk hormone receptor-positive breast cancers. The cytotoxic and antineoplastic effects of anthracyclines are mediated

through several mechanisms, such as DNA intercalation, topoisomerase II inhibition, and generation of reactive oxygen species (Venkatesh and Kasi, 2024). Anthracyclines like doxorubicin (DOX), an antibiotic derived from the *Streptomyces peucetius* bacterium, can significantly alter the gut microbiota (Gonçalves-Nobre et al, 2023; Johnson-Arbor and Dubey, 2024). DOX damages the intestinal lining, increasing permeability and the risk of endotoxemia (An et al, 2021). It also reduces microbial diversity and disrupts bacterial composition, increasing potentially harmful species while depleting beneficial bacteria (Gonçalves-Nobre et al, 2023). Although DOX disrupts the gut barrier similarly to other chemotherapies like cyclophosphamide, it results in less bacterial translocation to lymphoid organs, potentially due to its antimicrobial effects (Viaud et al, 2013).

Clinical studies also show that chemotherapy impacts microbial diversity and composition, which can influence immune response and tumor suppression. For instance, postmenopausal Dutch women with ER+ breast cancer treated with adjuvant or neoadjuvant chemotherapy (including DOX) had reduced α-diversity after treatment, with notable decreases in taxa like *Ruminococcaceae* and *Christensellaceae*, and an increase in *Lactobacillus* (Aarnoutse et al, 2021). In another pilot study, neoadjuvant chemotherapy increased gut diversity measures in breast cancer patients, an effect not seen with adjuvant chemotherapy (Wu et al, 2022). In preclinical studies, our group showed in a TNBC model that DOX responders had an elevated proportional abundance of *Akkermansia muciniphila* before DOX treatment compared to nonresponders. Further, DOX treatment increased *Akkermansia muciniphila*, reducing tumor weight and metastatic spread, underscoring the impact of gut microbiome changes on treatment outcomes (Bawaneh et al, 2022). These findings stand in contrast to those observed in breast cancer patients undergoing chemotherapy, who experienced a decrease proportional abundance of *Akkermansia*. Such discrepancies may arise from species-specific responses between mice and humans, as well as from treatment differences: Bawaneh et al focused on the effects of doxorubicin (DOX) alone, whereas Bilendike et al examined seven different chemotherapy combinations and sequences. Additional factors also play a role, including baseline microbiome differences and age, as breast cancer patients generally have distinct microbiomes compared to healthy individuals or preclinical models (Nandi et al, 2023; Peters et al, 2022). Host factors like diet, lifestyle, and genetic background further contribute to variability, as human patients have more diverse profiles than laboratory mice, which can influence how the microbiome responds to cancer treatments (Org and Lusis, 2018). Finally, differences in sample collection and analysis techniques could also account for these variations (Liu et al, 2024).

DOX is metabolized by certain gut bacteria, such as *Raoultella planticola*, which deglycosylates the drug into metabolites that may reduce its efficacy and toxicity (Yan et al, 2018). Further, gut microbiome depletion through antibiotics improved DOX efficacy in a TNBC murine model, suggesting that certain microbes may interfere with treatment and promote metastasis. For example, DOX treatment has been associated with increased *A. muciniphila*, which, through mucin degradation and SCFA production, can reinforce the gut barrier and reduce translocation of pathogens and small molecules. *A. muciniphila* also reduces inflammation by lowering levels of pro-inflammatory cytokines such as TNF-α, IL-6, and IL-12 (Ghotaslou et al, 2023).

Probiotics have shown the potential to enhance the efficacy of DOX in cancer treatment by influencing the gut microbiota and the tumor immune microenvironment. A study investigated three probiotic strains—*Bifidobacterium breve* BBr60, *Pediococcus pentosaceus* PP06, and *Bifidobacterium longum* BL21—demonstrating that while these probiotics alone did not directly induce antitumor effects, their combination with DOX resulted in a higher inhibition rate of tumor growth compared to DOX alone. This enhanced efficacy was attributed to the recruitment and infiltration of immune cells into the tumor region including increased infiltration of M1 macrophages and CD3 + CD8 + T cells, and elevated levels of cytokines IL-6, TNF-α, and IFN-γ. In addition, the probiotic *Bifidobacterium longum* BL21 increased the abundance of *Akkermansia*, which may help regulate the tumor microenvironment and improve immune responses. Thus, probiotics could be a valuable adjunct to chemotherapy by modulating gut microbiota and enhancing antitumor immunity, ultimately improving treatment outcomes for patients receiving DOX (Ye et al, 2024).

The CANTO prospective study found chemotherapy (anthracycline+taxane, taxane-based chemotherapy, and/or hormonotherapy) favored the colonization of favorable commensals such as *Methanobrevibacter smithii*, *D. formicigenerans*, *Blautia obeum*, or *R. torques*, which are known for their roles in maintaining gut health and supporting immune function. Conversely, chemotherapy was associated with a decrease in *Veillonella* species, which have been linked to worse prognostic outcomes in cancer patients (Terrisse et al, 2021). This dual effect suggests that chemotherapy not only disrupts the gut microbiome but can also create a more favorable microbial environment by promoting beneficial species. The presence of these favorable commensals may enhance treatment outcomes by improving the host's immune response and potentially mitigating some of the adverse effects associated with chemotherapy. For instance, both *B. obeum* and *R. torques* can produce SCFA that support intestinal health and modulate inflammation, which may help counteract some of the negative impacts of chemotherapy on the gut (Silva et al, 2020). Moreover, the study emphasizes the importance of understanding these microbiome changes as they could influence patient prognosis and treatment efficacy. By fostering a gut microbiome that is rich in beneficial bacteria, it may be possible to improve overall treatment outcomes for cancer patients undergoing chemotherapy. This highlights the potential for interventions such as probiotics or dietary modifications to support gut health during cancer treatment, ultimately aiming to enhance therapeutic efficacy and improve quality of life for patients. In summary, the findings from these clinical studies underscore the complex interplay between chemotherapy and the gut microbiome, suggesting that while chemotherapy induces dysbiosis, it can also promote beneficial microbial populations that may enhance patient outcomes. Further research into these relationships could lead to innovative strategies for optimizing cancer treatment through microbiome modulation.

## Antimetabolite chemotherapy

Antimetabolites, such as capecitabine and gemcitabine, function as structural analogs of DNA or RNA components, inserting themselves in place of nucleotides during replication. This interruption inhibits DNA synthesis and ultimately causes cell death (Chandraprasad et al, 2022). Studies indicate these drugs can affect the gut microbiome and the gut microbiome can also affect

the efficacy of these drugs (Li et al, 2024). A study examining the gut microbiota of HER2-negative metastatic breast cancer patients found distinct differences in composition and function between those receiving metronomic chemotherapy with Capecitabine ($n = 15$)—a low, minimally toxic dose—and those on a conventional chemotherapy regimen ($n = 16$). The metronomic group showed reduced gut microbiota diversity compared to the conventional group. At the phylum level, patients in both dosage groups showed similar microbial compositions, with Bacteroidetes, Firmicutes, Proteobacteria, and Actinobacteria making up over 95% of the microbiome. *Megamonas* and *f_Mogibacteriaceae* were enriched in the metronomic group, while *Blautia* and *o_Streptophyta* were reduced. Notably, in patients receiving metronomic chemotherapy, higher abundance of *Blautia obeum* was associated with significantly longer progression-free survival (PFS; 32.7 vs. 12.9 months, $P = 0.013$), whereas the presence of *Slackia* correlated with shorter PFS (9.2 vs. 32.7 months, $P = 0.004$). *B. obeum* is implicated in the transformation of carcinogenic heterocyclic amines, with decreased levels potentially increasing cancer risk. Conversely, *Slackia* has been linked to colorectal cancer development in several studies, indicating its potential as a microbial biomarker for cancer prevention, diagnosis, and treatment (Guan et al, 2020).

Gemcitabine (2′,2′-difluorodeoxycytidine), a deoxycytidine analog that inhibits DNA synthesis, is used to treat advanced or metastatic breast cancer (Ferrazzi and Stievano, 2006). The antitumor activity of gemcitabine depends on its activation or degradation, where cytidine deaminase plays a pivotal role in the degradation process. Bacteria, mainly belonging to the class Gammaproteobacteria and *Mycoplasma hyorhinis*, can metabolize gemcitabine in its inactive form (2′,2′-difluorodeoxyuridine) by the bacterial enzyme cytidine deaminase (Geller et al, 2017; Vande Voorde et al, 2014). In a colon cancer mouse model, gemcitabine resistance caused by intratumoral Gammaproteobacteria was reversed by co-administration of ciprofloxacin antibiotic, thus, supporting specific bacteria might contribute to drug resistance (Geller et al, 2017).

### Platinum-based chemotherapy

Platinum compounds induce DNA double-strand breaks by forming intra-strand platinum-DNA adducts, thereby inhibiting DNA replication. Therefore, platinum compounds are extremely toxic to the rapidly dividing cells of the intestinal mucosa, resulting in impaired barrier function. Disruption in intestinal barrier integrity allows microbes to enter circulation, potentially contributing to systemic inflammation (Iida et al, 2013). Interestingly, tumor-bearing mice that are either germ-free or treated with broad-spectrum antibiotics show a diminished response to oxaliplatin or cisplatin. This reduced efficacy is linked to the depletion of microbes, leading to downregulated expression of genes associated with antigen presentation, inflammation, phagocytosis, and adaptive immunity, while genes related to cancer progression and metabolism are upregulated (Iida et al, 2013). In addition, the depletion of microbiota inhibited the generation of reactive oxygen species (ROS) by tumor-infiltrating myeloid cells. This inhibition is significant because ROS plays a crucial role in platinum drug-induced cellular damage, including DNA damage and the initiation of apoptotic responses in tumor cells (Iida et al, 2013). The SCFA butyrate enhances platinum compound

oxaliplatin's therapeutic effect by inhibiting histone deacetylases (HDACs) and inducing the expression of inhibitor of DNA binding 2 (ID2). Consequently, ID2 inhibits E2A and induces the expression of IL-12 receptors on the surface of CD8 + T cells, enhancing IL-12 signaling and improving the antitumor ability of CD8 + T cells (Danne and Sokol, 2021). The interaction between platinum-based chemotherapy and the gut microbiome is multifaceted, affecting both treatment efficacy and patient quality of life. Maintaining a healthy gut microbiome may enhance therapeutic responses while mitigating side effects.

## Immunotherapy

Immunotherapy has emerged as a promising advancement in breast cancer treatment, especially for TNBC. Although breast cancer was historically considered poorly immunogenic and not extensively studied for its potential response to immunotherapy, recent successes with immune checkpoint blockade in other cancers, along with growing evidence of the immune system's role in cancer progression, have prompted the development of clinical trials exploring various immunotherapy strategies for breast cancer patients (Debien et al, 2023). For example, pembrolizumab (an anti-programmed cell death protein 1 (PD-1) antibody) has been approved in combination with chemotherapy to treat both early-stage and metastatic triple-negative breast cancer (Cortes et al, 2022; Schmid et al, 2020). However, immunotherapy efficacy varies among individuals and may be accompanied by immune-related adverse events. The gut microbiome is now well acknowledged for its critical role in immunotherapy in other cancer types, with preclinical studies using mouse models showing that specific gut microbial populations can affect the response to immunotherapy (Choi et al, 2023; Zhang et al, 2023).

Gut microbiota mediates the response to anti-PD-1 and programmed death ligand 1 (PD-L1) immunotherapies, which blocks the interaction between PD-1 on T cells and PD-L1 on tumor cells or antigen-presenting cells (Lu et al, 2022). This blockade prevents the inhibition of T-cell activity, allowing T cells to remain activated and attack tumor cells (Javed et al, 2024). Gut commensals *B. longum*, *Collinsella aerofaciens*, and *E. faecium* are associated with anti-PD-1 cancer efficacy (Matson et al, 2018). *B. longum* increased anti-PD-1 therapy in a murine model of TNBC (Kim et al, 2021). A better response to anti-PD-L1 therapy was also observed in mice with specific species of microbiota, including *A. muciniphila*, *B. longum*, *Collinsella aerofaciens*, and *Faecalibacterium prausnitzii* (Alpuim Costa et al, 2021). Administering *Akkermansia* increased tumor-infiltrating lymphocytes (TILs) and improved ICB efficacy in germ-free mice following microbial transplant (Routy et al, 2018). In a TNBC murine model, anti-PD-1 therapy increased *Akkermansia* abundance (Pingili et al, 2021). Clinical evidence highlights the role of the microbiome in ICB efficacy, with responders showing an increased abundance of key bacterial species, including *Akkermansia* and *Alistipes* (Routy et al, 2018). Increased levels of *A. muciniphila* in responders to anti-PD-L1 therapy are associated with greater activation of dendritic cells, leading to increased IL-12 secretion, enhanced migration of CD4 + CCR9+ memory T cells, and CD4 + CXCR3 + T cells from mesenteric lymph nodes to tumor-draining lymph nodes, thereby increasing antitumor responses (Zhou et al, 2022).

Patients who responded to anti-PD-1 therapy displayed greater microbial diversity and specific enrichment of *Clostridiales*, *Ruminococcaceae*, and *Faecalibacterium* (Matson et al, 2018). *Ruminococcaceae*, *Clostridiales*, and *Faecalibacterium* contribute to antitumor effects by increasing the ratio of CD4+ to CD8 + T cells while reducing the activity of regulatory T cells and myeloid-derived suppressor cells. Both bacteria and their metabolites may influence anti-PD-1/PD-L1 therapy outcomes by promoting Th1 cell differentiation, enhancing DC function, and reducing circulating regulatory T cells (Tregs). These effects help reduce immunosuppression and strengthen immune activation (Zhou et al, 2022).

In a clinical trial comparing outcomes in patients with hormone receptor-positive metastatic breast cancer who received eribulin (antitubulin antimitotic agent) with or without pembrolizumab, a shift in the abundance of *Akkermansia* and *Faecalibacterium* after two cycles of therapy (C2) were observed. For patients receiving eribulin who had a partial response, the levels of *Faecalibacterium* increased significantly from 4.3% at baseline to 13.9% after C2. However, this change wasn't seen in patients with stable disease. In the group receiving eribulin and pembrolizumab, these shifts weren't observed. Patients with progression-free survival (PFS) below the median had a decrease in *Akkermansia* levels from 5.7% at baseline to less than 1% after C2. Those with overall survival (OS) above the median saw a decrease in *Akkermansia* from 5.3% at baseline to 2.4% after the C2, while those with OS below the median had similar alpha-diversity scores at baseline compared to those with longer OS in both treatment groups (de Sousa et al, 2020). Although, it is important to note that ICB is not currently approved for HR+ breast cancer. Another clinical trial (NCT03586297) is currently investigating the association between gut and intratumoral microbiota with pathologic response in newly diagnosed TNBC patients undergoing neoadjuvant chemotherapy.

Bacterial metabolites that are derived from fiber fermentation, bile acid metabolism, lipid metabolism, and cholesterol metabolism/elimination interfere directly or indirectly with tumor cell proliferation and differentiation and, therefore, can enhance therapeutic efficacy (Di Modica et al, 2022). Microbial-derived trimethylamine *N*-oxide (TMAO) correlates with improved efficacy of immunotherapy in TNBC by inhibiting tumor growth through activation of CD8 + T-cell-mediated antitumor immunity. TMAO also activates antitumor immunity by inducing pyroptosis of tumor cells (Liu et al, 2023b). Intestinal *B. pseudopodium* modulated enhanced immunotherapy response through the production of the metabolite inosine (Mager et al, 2020). Synthetic strains of *Escherichia coli* engineered to produce high levels of L-arginine can colonize tumors and enhance the anticancer efficacy of anti-PD-L1 antibodies by enhancing the function of effector and memory T cells (Canale et al, 2021). Similarly, other microbial interventions, including probiotics, have been explored for their potential to enhance the effectiveness of tumor immunotherapies.

Probiotics have been shown to enhance tumor suppression when combined with tumor immunotherapy. They may help restore microbial diversity, which is often reduced in cancer patients, potentially improving the effectiveness of immune checkpoint blockade (ICB) treatments (Singh et al, 2023). Probiotics can modulate the immune system by boosting the activity of immune cells such as natural killer (NK) cells and T cells, which are critical for effective antitumor responses. For instance, certain probiotic strains, such as *Lactobacillus helveticus* and *Lactobacillus plantarum*, have been shown to increase NK cell activity and promote a Th1-biased immune response, both of which are linked to better tumor control (Malik et al, 2018). In a mouse melanoma model, oral administration of *Bifidobacterium* spp. improved the efficacy of PD-L1 inhibitors and nearly eliminated tumor growth by activating dendritic cells and enhancing tumor-specific CD8 + T cells (Sivan et al, 2015). However, clinical studies have also indicated that patient use of probiotics may be associated with poorer responses to ICB in melanoma. These discrepancies may be due to the broad classification of probiotics, as different species may produce varying effects on outcomes (Spencer et al, 2021). Therefore, additional studies are needed to better understand the impact of combining probiotics with ICB in the context of breast cancer development.

## Endocrine-targeting therapies

Hormone receptor-positive (HR +) breast cancer, which accounts for ~70–80% of all diagnosed cases in pre-menopausal (60%) and postmenopausal (75%) women, is primarily treated with endocrine-targeting therapies (Ferreira Almeida et al, 2020; Lumachi et al, 2015). Endocrine-targeting therapies, like estrogen receptor-α (ER) blockade or aromatase inhibitors (AI), either block ER activity or suppress estrogen biosynthesis, inhibiting the growth of ER+ breast cancer cells (Terrisse et al, 2023). The main types of therapies used for ER+ breast cancer include selective estrogen receptor modulators (SERMs), such as tamoxifen (TAM), selective estrogen receptor degraders (SERDs), such as fulvestrant, or AIs, such as letrozole, anastrozole, and exemestane.

The gut microbes' role in HR+ breast cancer responses to endocrine-targeting therapies remains elusive as there are currently no systematic studies on the interplay between the gut microbiome and endocrine-targeting therapy efficacy. One key function of the gut microbiota is its influence on estrogen metabolism. Gut bacteria significantly affect enterohepatic estrogen circulation through β-glucuronidase (GUS) enzymes to deconjugate estrogen from its inactive, conjugated form, facilitating its reabsorption in the intestines rather than excretion in feces (Baker et al, 2017; Plottel and Blaser, 2011). Only about 7% of conjugated estrogen metabolites are excreted (Gorbach, 1984); most is hydrolyzed back to its active form and reabsorbed, a process involving GUS-producing bacteria like *Alistipes, Bacteroides*, and *Lactobacillus* species (Edwinson et al, 2022; Terrisse et al, 2023). This GUS-mediated deconjugation plays a significant role in the microbiome's influence on estrogen levels, and dysregulation of this process has been linked to microbial imbalances.

Specifically, estrogen deconjugation by GUS enzymes has been linked to microbial dysbiosis in women with breast cancer. Notably, breast cancer patients have elevated GUS levels in nipple aspirate fluid compared to healthy women (Nandi et al, 2023). Research in healthy postmenopausal women found that greater microbial diversity in the gut was positively associated with higher ratios of estrogen metabolites to parent estrogens (Kwa et al, 2016). In mice, long-term estrogen supplementation altered the gut microbiota by decreasing *A. muciniphila* levels and reducing GUS activity, suggesting that changes in estrogen levels could influence gut microbiota composition (Chen et al, 2018b). Studies also indicate

that lower bacterial biodiversity in stool samples is linked to increased estrogen excretion and a higher risk of breast cancer (Flores et al, 2012). Although not statistically significant, postmenopausal women with breast cancer have less diverse and different gut microbiota compared to age-matched healthy women, alongside higher systemic estrogen levels (Goedert et al, 2015). Together, these findings suggest that variations in gut microbial diversity may impact estrogen metabolism and circulating estrogen levels (Fuhrman et al, 2014).

These shifts in microbial composition may therefore have broader implications beyond estrogen metabolism, particularly in the context of cancer treatment. Bacterial GUS may limit the efficacy of cancer therapeutics by reversing the glucuronidation process, which inactivates and detoxifies drugs and other molecules by increasing their water solubility to promote their removal through the kidneys or gastrointestinal (GI) tract (Dutton, 2019). GUS bacteria remove the glucuronic acid moiety, regenerating the original molecule marked initially for elimination in the gut (Pellock and Redinbo, 2017). The anticancer prodrug irinotecan, used to treat colorectal and pancreatic cancers, is the archetype of bacterial GUS-induced drug toxicity. SN-38, the active form of irinotecan, is inactivated in the liver via glucuronidation, which is reversed by GUS bacteria, reforming SN-38, which causes GI toxicities, including diarrhea (Pellock and Redinbo, 2017). Inhibitors for bacterial GUS alleviated SN-38 toxicity in mice (Wallace et al, 2010). Using an in vitro model, Chen et al found tamoxifen and 4-hydroxytamoxifen, the active metabolite of tamoxifen, are potential GUS bacteria substrates as well (Chen et al, 2018a). However, further research is needed to understand how gut microbiota may metabolize or alter oral endocrine-targeting therapies, potentially influencing their efficacy and affecting cancer recurrence.

Understanding these mechanisms is critical, as changes in drug metabolism by the gut microbiota could intersect with its broader role in metabolic processes and disease outcomes. For instance, while the gut microbiome influences sex hormone homeostasis, it also reacts to alterations in estrogen status, which can contribute to metabolic dysfunction and disease progression (Cross et al, 2024). Cross et al demonstrated that ovariectomy increased gut permeability and inflammation in mice, with a high-fat diet exacerbating these effects. In mice fed a low-fat diet (but not a high-fat diet), ovariectomy also altered the gut microbiome, leading to increased fecal GUS activity. When fecal microbiota from low-fat diet-fed ovariectomized mice were transplanted into gnotobiotic mice, the recipients gained more weight and exhibited higher expression of genes associated with metabolic dysfunction and inflammation compared to those receiving microbiota from ovary-intact control mice (Cross et al, 2024). These findings suggest that the gut microbiome is responsive to estrogen fluctuations and plays a significant role in metabolic issues, potentially influencing breast cancer risk.

The gut microbiome's influence on estrogen signaling could have implications for the response to targeted therapies in breast cancer. HDAC inhibitors are being developed for the treatment of breast cancer as HDAC inhibitors induce ERα degradation in breast cancer cells. Interestingly, SCFAs, such as butyrate, propionate, and acetate, produced in the gut through microbial degradation of high-fiber diets, also exhibit activity as HDAC inhibitors (Schoeller et al, 2022). Schoeller et al demonstrated that

SCFAs induce degradation of both wild-type and mutant ERα in MCF-7 and T47D ER+ breast cancer cells. Butyrate, in particular, inhibited tumor growth and downregulated mutant ERα in an athymic nude mouse orthotopic model bearing MCF-7-ERα-Y537S cells (Schoeller et al, 2022). These results suggest that microbial-derived metabolites, such as SCFAs, could serve as novel dietary interventions to enhance the effectiveness of current therapies for ER+ breast cancer.

Building on these insights, our group is currently investigating the effects of oral aromatase inhibitor (AI) treatment on gut microbiota shifts in breast cancer patients (NCT05030038). Results from this clinical trial will reveal how AI treatment impacts gut microbial diversity, bacterial-derived bioactive compounds, and glucuronidated sex hormones, potentially influencing breast cancer treatment outcomes. Previous studies have shown that lower microbial diversity, which is associated with higher levels of circulating estrogens, correlates with a higher risk of developing breast cancer (Chen and Madak-Erdogan, 2016; Flores et al, 2012). Future research should explore the interplay between oral endocrine therapies, such as tamoxifen, and the gut microbiota in ER+ breast cancer, particularly how microbial metabolism of these therapies may affect their efficacy and influence cancer recurrence.

## Other targeted therapies

HER2 is amplified or overexpressed in 20–30% of all breast cancer patients and is sensitive to HER2 inhibitors, such as trastuzumab, pertuzumab, trastuzumab emtansine, and fam-trastuzumab deruxtecan (Alpuim Costa et al, 2021; Dumbrava et al, 2019). In a study of HER2-positive breast cancer patients who were treated with neoadjuvant Trastuzumab, gut α-diversity measures assessed before treatment showed significantly higher α-diversity in responders ($n = 16$) than in nonresponders ($n = 7$). Responders also showed a microbiota that was enriched in bacteria from the Clostridiales (i.e., Lachnospiraceae), Bifidobacteriaceae, Turicibacteriaceae, and Prevotellaceae taxa, whereas Bacteroides commensals were more abundant in nonresponders (Di Modica, 2020). In a preclinical mouse model, the antitumor activity of trastuzumab was reduced by antibiotic treatment or FMT from antibiotic-treated donors. However, FMT from trastuzumab-responding patients improved response in mice, while FMT from non-responding patients did not, suggesting that specific microbial compositions can enhance or reduce the efficacy of HER2-targeted therapies. This microbiota modulation was reflected in tumors by reduced recruitment of CD4 + T cells and granzyme B–positive cells after trastuzumab treatment. However, administering Lactococcus lactis or Lactobacillus paracasei with trastuzumab improved its efficacy in mice under vancomycin regimens (Di Modica, 2020).

The combination of cyclin-dependent kinase (CDK) 4/6 inhibitors with endocrine therapy (ET), either aromatase inhibitor (AI) or fulvestrant, is used to treat HR + /HER2-negative metastatic breast cancer (MBC). In a study of 14 MBC patients treated with either palbociclib, ribociclib, or abemaciclib with an AI or fulvestrant, gut microbiota and therapeutic efficacy were assessed, and patients were classified as responders (R) and nonresponders (NR) according to progression-free survival. While

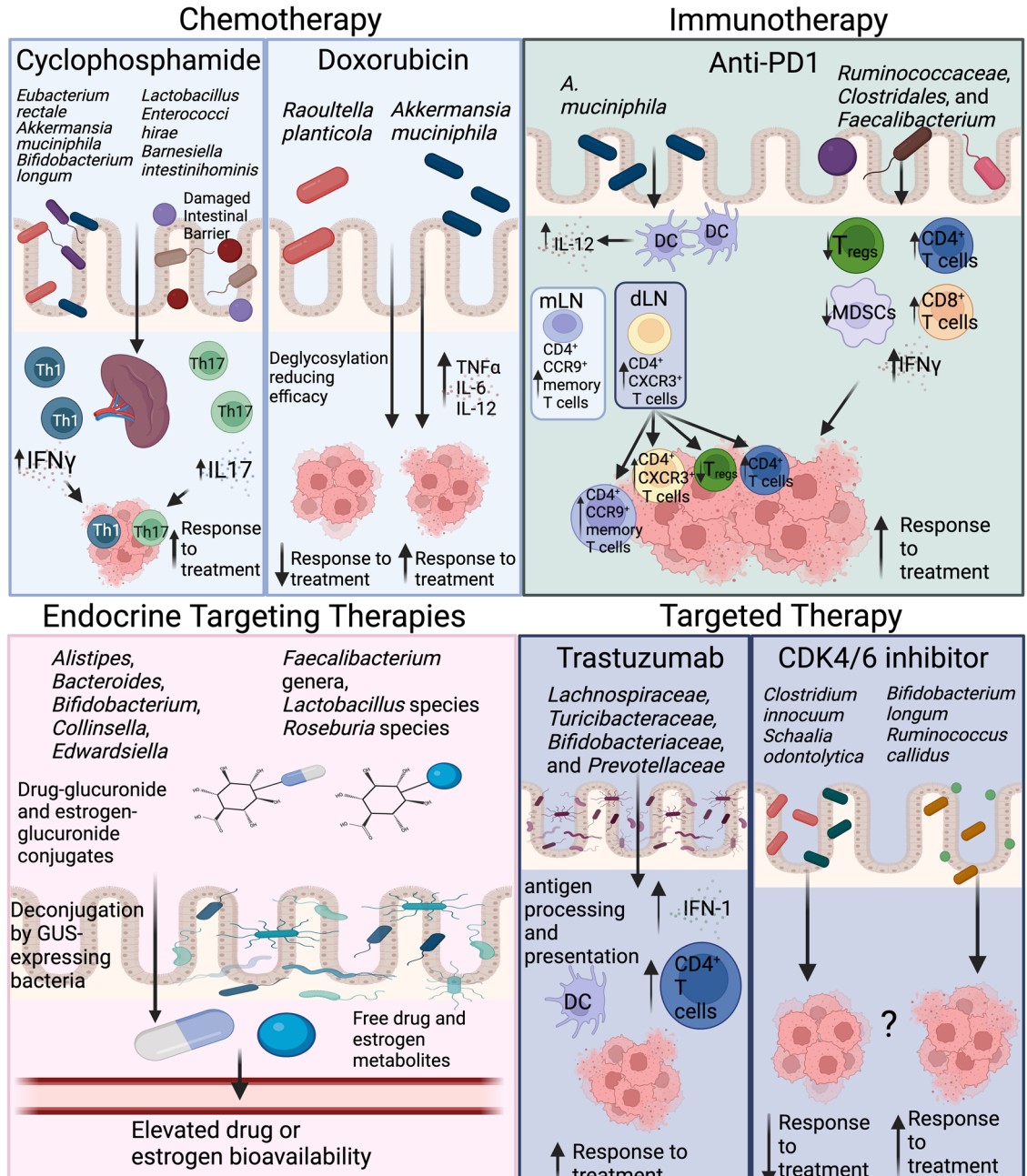

**Figure 1. Reported gut microbiome-host interactions associated with breast cancer therapeutic modalities.**

Adapted from published data (Arnone and Cook, 2022; Di Modica et al, 2022; Schettini et al, 2023; Zhao et al, 2023; Zhou et al, 2022). Th1 T helper type 1 cell, TH17 T helper 17 cells, IFN-γ interferon-gamma, IL-17 interleukin-17, TNF-α tumor necrosis factor alpha, IL-6 interleukin-6, IL-12 interleukin-12, DC dendritic cells, mLN mesentery lymph nodes, dLN tumor-draining lymph nodes, Tregs regulatory T cells, MDSCs myeloid-derived suppressor cells, GUS beta-glucuronidase. Created in https://BioRender.com.

there were no significant differences observed between R and NR in terms of α-/β-diversity at the phylum and species level, four bacterial species were identified as distinguishing markers: *B. longum* and *Ruminococcus callidus* were associated with R, while *Clostridium innocuum* and *Schaalia odontolytica* were linked to NR. Further, network analysis revealed two main clusters of interacting bacterial species, or species Interacting Groups (SIGs).

The SIG1 cluster contained 75% of species linked to NR, while the SIG2 cluster had 76% of species more abundant in R. This statistically significant distribution suggests that these two groups of bacteria may play opposite roles in how well patients respond to CDK4/6 inhibitors (Schettini et al, 2023). The potential impact of the gut microbiome on breast cancer therapeutic responsiveness is overviewed in Fig. 1.

## Obesity-induced gut dysbiosis and breast cancer therapies

Obesity negatively affects various aspects of breast cancer treatment, including surgical outcomes, chemotherapy and endocrine-targeting therapy effectiveness, and recurrence risk (Harborg et al, 2023). The link between obesity and breast cancer risk and prognosis is driven by altered drug metabolism, increased inflammation, and changes in the tumor microenvironment (Lee et al, 2019). These mechanisms underlying cancer risk can be influenced by the gut microbiome, potentially affecting cancer progression and treatment response (Arnone et al, 2024). Animal studies have shown that gut dysbiosis disrupts immune function, promoting the initiation and progression of breast tumors. Notably, our group has shown that ablation of the gut microbiome through antibiotic administration increased tumor-free survival in both low-fat and high-fat lard diet-fed mice (Soto-Pantoja et al, 2021). Further, mice fed a lard-based diet or mice receiving FMT from lard-fed donors had reduced tumor-free survival rates compared to low-fat control diet-fed mice. However, tumor-free survival improved to 50% in lard-fed mice receiving transplants from low-fat control diet-fed donors. Lard-fed mice also showed greater tumor weight and multiplicity than low-fat controls, with similar effects observed in low-fat control diet-fed mice receiving transplants from lard-fed donors. In addition, tumor latency was significantly shorter in lard-fed mice, implicating diet-modulated gut microbiota in breast cancer risk (Soto-Pantoja et al, 2021). Similarly, high-fat diet-derived FMT in mice on a low-fat diet reduced the effectiveness of chemotherapy, as evidenced by increased tumor size, weight, and lung metastases compared to DOX-responding mice. Tumors from high-fat FMT + DOX-treated mice showed decreased apoptosis (reduced cleaved caspase 3), suggesting that the high-fat diet FMT hindered DOX-induced tumor cell death in a 4T1 syngeneic model. Additionally, these mice had fewer macrophages in their tumors. Metagenomic sequencing revealed that high-fat diet FMT altered the gut microbiota, increasing the abundance of several bacterial species, including *Enterorhabdus caecimuris*, which was associated with the DOX nonresponder phenotype (Bawaneh et al, 2022).

Obesity-related gut dysbiosis promotes chronic low-grade inflammation, which contributes to cancer progression and reduces the effectiveness of treatments like chemotherapy and immunotherapy by altering the tumor microenvironment (Avtanski et al, 2023). Dysbiosis also impacts immune cell composition within the microenvironment, such as decreasing tissue-resident memory T cells, as observed in gastric cancer (Cao et al, 2024). Notably, in a TNBC murine model, anti-PD-1 treatment effectively reduced obesity-driven tumor progression. This response may be linked to an increased presence of antitumor immune cells, including CD8 + T cells and NK cells, in tumor-adjacent adipose tissue, alongside reduced inflammatory signaling (Pingili et al, 2021).

In a preclinical obesity model, breast cancer progression and anti-PD-L1 therapy were studied following weight loss induced by either vertical sleeve gastrectomy (VSG) or diet. Both approaches protected against obesity-driven tumor progression, but diet was more effective than VSG in reducing tumor burden despite achieving similar weight loss. VSG tumors exhibited heightened inflammation, increased PD-L1+ immune and non-immune cells, reduced T lymphocytes, and lower cytolytic markers, signifying a less effective antitumor microenvironment. While obese mice were generally resistant to immune checkpoint blockade, anti-PD-L1 therapy significantly impaired tumor progression in VSG-treated mice by boosting antitumor immunity. These findings highlight the conserved mechanisms by which obesity and bariatric surgery-induced weight loss influence breast cancer outcomes (Sipe et al, 2022). Collectively, these findings underscore the critical role of obesity on the gut microbiome in modulating immune function, inflammatory responses, and drug metabolism, highlighting its potential as a target for enhancing breast cancer treatment efficacy and reducing the negative impact of obesity on therapeutic outcomes.

## Clinical perspective

The gut microbiome is involved in a complex network of innate and adaptive immune responses, with increasing evidence linking differences in gut microbiome composition with cancer pathogenesis and outcomes. The influence of the gut microbiome in breast cancer pathogenesis, treatment response, and outcomes highlights its potential importance in clinical trial design and clinical practice. Incorporating microbiome analysis into breast cancer clinical trials can refine our understanding of how gut bacteria affect drug metabolism, therapeutic efficacy, treatment-related side effects, and disease outcomes. For instance, future studies could investigate how specific microbial profiles or shifts in gut diversity influence responses to hormone therapies like aromatase inhibitors or chemotherapeutic agents such as doxorubicin and cyclophosphamide. By evaluating the microbiome's role in modulating drug metabolism and immune responses, researchers could identify biomarkers for predicting treatment outcomes and adverse effects. Microbiome interventions, including FMT, probiotics, and dietary modifications, can potentially boost immune checkpoint inhibitor response rates by promoting favorable microbial compositions. These approaches may also reduce immune-related side effects, such as colitis, by modulating inflammatory responses. Additionally, microbiome-based strategies could help overcome therapy resistance, with studies indicating FMT and probiotics may re-sensitize resistant tumors. Personalized microbiome profiling could further tailor these interventions, maximizing benefits while enhancing chemotherapy's efficacy and reducing its toxicity.

Diet assessment is critically underutilized in clinical practice, despite its potential to significantly influence drug metabolism and gut microbiome composition (Whelan et al, 2024). Dietary patterns can rapidly and substantially alter microbial populations, potentially affecting drug absorption, efficacy, and interactions (Whelan et al, 2024). While healthcare professionals recognize the importance of understanding food-drug interactions, there remains a notable lack of standardized assessment tools and comprehensive knowledge about how specific dietary components might modify drug responses (Osuala et al, 2021). The complex interplay between diet, gut microbiome, and pharmacokinetics represents a frontier in personalized medicine, highlighting the need for more sophisticated approaches to patient care that consider individual dietary habits as a crucial factor in treatment outcomes. This gap in current clinical practice suggests an urgent need for research developing practical, comprehensive methods to assess and integrate dietary information into medication management strategies, ultimately

Table 1. Current clinical trials investigating the interactions between gut microbiome and breast cancer therapies.

| Title | Clinical trial no. | Study design | Objective | Intervention | Status |
|---|---|---|---|---|---|
| Intestinal microbiota impact for prognosis and treatment outcomes in early luminal breast cancer and pancreatic cancer patients | NCT05580887 | Observational model: Cohort | To identify GM patterns associated with poor and favorable treatment outcomes in breast cancer and pancreatic cancer patients | mFOLFIRINOX, Doxorubicin, Cyclophosphamide, Paclitaxel, Carboplatin | Recruiting |
| Gut microbiome components predict response to neoadjuvant therapy in HER2-positive breast cancer patients: a prospective study | NCT05444647 | Observational model: Cohort | To determine the characteristics and alterations of the gut microbiome during neoadjuvant therapy for HER2-positive breast cancer patients, as well as the relation between the gut microbiome and the probability of pCR. | Trastuzumab | Recruiting |
| Association between changes in the gut microbiome and chemotherapy-induced nausea in stage I-III breast cancer | NCT05417867 | Observational model: Cohort | To understand how changes in the bacteria composition (microbiome) of the gut may be associated with the occurrence of chemotherapy-induced nausea (CIN) in women undergoing chemotherapy for stage I-III breast cancer. | Chemotherapy | Recruiting |
| The gut microbiome and immune checkpoint inhibitor therapy in solid tumors | NCT05037825 | Observational model: Cohort | To assess the associations between the gut microbiota (composition and function), host immune system, and ICI treatment efficacy across multiple cancer types. | Anti-PD-1, anti-PD-L1, or anti-CTLA-4 as a single agent or in combination with another checkpoint inhibitor or other treatment agent or modality (e.g., targeted therapy, chemotherapy, surgery, radiation, etc.) | Recruiting |
| The intestinal microbiome in triple-negative breast cancer treated with immunotherapy (IMPACT) | NCT06318507 | Observational model: Cohort | To determine how the intestinal microbiome differs between patients with obesity and early triple-negative breast cancer who achieve a pathologic complete response from preoperative anti-PD-1 immunotherapy (pembrolizumab) versus patients who do not. | Pembrolizumab | Recruiting |
| Relationship between alterations in the GI microbiome and GI inflammation on symptom burden in women with breast cancer receiving chemotherapy | NCT06238986 | Observational model: Cohort | To evaluate the relationship between alterations in the GI microbiome and GI inflammation on symptom burden in women with breast cancer receiving chemotherapy. | Chemotherapy (Taxotere + cyclophosphamide treatment +/- trastuzumab) | Recruiting |
| A study of the gut microbiome in hormone receptor-positive HER2-negative breast cancer treated with CDK4/6 inhibitors | NCT06171360 | Observational model: Cohort | To determine the interplay between the gut microbiome (its composition and evolution during treatment), circulating immune, metabolic, and cytokine biomarkers (before and during treatment), and response outcomes to the CDK4/6 inhibitor. | CDK4/6 inhibitor | Recruiting |
| Gut and intratumoral microbiome effect on the neoadjuvant chemotherapy-induced immunosurveillance in triple-negative breast cancer | NCT03586297 | Observational model: Cohort | To correlate gut and intratumoral microbiome composition and antitumor immune responses with pCR in newly diagnosed triple-negative breast cancer (TNBC) patients undergoing standard-of-care neoadjuvant chemotherapy | Chemotherapy | Recruiting |
| Oral aromatase inhibitors modify the gut microbiome | NCT05030038 | Observational model: Cohort | To describe the shift of gut microbiome from baseline after 4-weeks and 12-weeks of oral aromatase inhibitor treatment. | Aromatase Inhibitor | Recruiting |

**Table 1.** (continued)

| Title | Clinical trial no. | Study design | Objective | Intervention | Status |
|---|---|---|---|---|---|
| Abemaciclib in treating patients with surgically resectable, chemotherapy-resistant, triple-negative breast cancer | NCT03979508 | Interventional | To explore putative mechanistic connections underlying bacteria-drug interactions in all patients and attempt to identify the biomolecular features within the gut (stool) microbiome and its association with the pharmacokinetics and pharmacodynamics of abemaciclib. | Abemaciclib | Recruiting |
| Rifaximin for the treatment of gastrointestinal toxicities related to pertuzumab-based therapy in patients with stage I–III HER2-positive breast cancer | NCT04249622 | Interventional | Evaluate changes in the fecal microbiome, hydrogen breath test, and permeability test before and after pertuzumab-based chemotherapy. | Pertuzumab | Active, not yet recruiting |
| Assessing the impact of the microbiome on breast cancer radiotherapy toxicity | NCT04245150 | Observational model: Cohort | To determine if side effects from breast radiation are associated with gut microorganisms. | Radiotherapy | Active, not yet recruiting |
| Real-world study of pyrotinib in neoadjuvant therapy for HER2-positive breast cancer | NCT05561686 | Observational model: Cohort | To explore the efficacy and safety of pyrotinib-based neoadjuvant therapy for HER2-positive early or locally advanced breast cancer patients; exploratory analysis to explore the correlation between TMB levels and pCR rate of neoadjuvant therapy in HER2-positive breast cancer patients, and the effect of pyrotinib-based neoadjuvant therapy on intestinal flora. | Pyrotinib | Not yet recruiting |
| Neoadjuvant treatment of locally advanced breast cancer patients with ribociclib and letrozole | NCT05163106 | Interventional | To improve understanding of tumor responses and resistance in patients suffering from ER-positive/HER2 negative locally advanced breast cancer, focusing on the role of the immune system including the gut microbiome. | Letrozole 2.5 mg oral tablet; Ribociclib 600 mg oral tablet Goserelin | Completed |
| Effect of radiotherapy variables on circulating effectors of immune response and local microbiome | NCT03383107 | Observational model: Other | To study microbial changes and how these changes correlate with alteration in immune mediators (i.e., lymphocytes, cytokines) present in blood samples before, during and after radiation; and explore the association between these parameters and type of radiation received. | Radiotherapy | Completed |
| Determinants of acquired endocrine resistance in metastatic breast cancer: a pilot study (ENDO-RESIST) | NCT04579484 | Observational model: Cohort | To identify markers of endocrine resistance in ctDNA and the gut microbiome in patients with ER + HER2- metastatic breast cancer. | Aromatase Inhibitor and a CDK4/6 inhibitor | Unknown |
| Intestinal microbiota of breast cancer patients undergoing chemotherapy | NCT04138979 | Observational model: Case-control | To determine changes in gut microbiota before and after chemotherapy treatment | Cyclophosphamide | Unknown |

aiming to optimize treatment efficacy and minimize potential adverse interactions.

Currently, several ongoing clinical trials are investigating the relationship between gut microbiota and breast cancer, summarized in Table 1. These trials aim to elucidate the complex interactions between the gut microbiome and breast cancer, potentially leading to new therapeutic strategies and improved patient outcomes. These clinical trials focus on several key areas: the relationship between gut microbiota composition and chemotherapy response, the impact of probiotics on immune responses in breast cancer patients, changes in the gut microbiome during and after cancer treatment, and the potential of gut microbiota as a biomarker for breast cancer risk and prognosis. Clinical trials could also integrate microbiome profiling to stratify patients based on their gut microbiota composition, allowing for more personalized and potentially more effective treatment strategies. In addition, as knowledge is gained regarding the microbiome and breast cancer risk, methods to alter a host's microbiome could be explored as prevention.

## Current limitations

While animal models provide valuable insights, the human gut microbiome is considerably more complex and diverse, shaped by unique diets, environments, and physiological factors that may limit the direct applicability of animal findings (Feng et al, 2024; Sundberg and Schofield, 2018). Studies show that breast cancer patients often have altered gut microbiota, though the specific changes and impacts vary widely among individuals, making it challenging to develop universal microbiome-based interventions (Nandi et al, 2023). Many animal studies focus on short-term outcomes, while breast cancer treatments in humans typically last months or years, leaving the long-term effects of microbiome modulation on treatment efficacy and side effects largely unknown. Although links between gut microbiota and treatment outcomes have been observed, the exact mechanisms by which gut bacteria influence breast cancer progression and response to treatment in humans are not yet fully understood. Additionally, examining the functional outputs of gut microbiota, such as differences in microbial genes involved in fiber metabolism, and metabolite profiles can provide deeper insights into how specific microbial activities influence clinical outcomes and enhance our ability to identify meaningful correlations across multiple studies. To bridge this gap, future research should prioritize larger, longitudinal studies in human breast cancer patients to monitor microbiome changes during treatment, develop personalized microbiome interventions based on patient profiles, investigate the safety and efficacy of microbiome-modulating therapies in clinical trials, and clarify the specific mechanisms by which gut bacteria impact breast cancer and treatment response. Expanding our knowledge of the gut microbiome's role in breast cancer could pave the way for more effective, microbiome-informed therapies for patients.

## Conclusions

The gut microbiome has been shown to influence the efficacy of breast cancer therapies by affecting drug metabolism, immune responses, and hormone regulation. Modulating the gut microbiome through probiotics, prebiotics, and dietary interventions may enhance the efficacy of breast cancer treatments by promoting a more favorable microbial environment (Nandi et al, 2023). Further, obesity-induced gut dysbiosis is linked to resistance to breast cancer therapies through altered drug metabolism and immune responses. Interventions targeting the gut microbiome may offer promising strategies to overcome this resistance and improve treatment outcomes. Further, maintaining gut homeostasis through a balanced diet rich in fiber, using probiotics or prebiotics, and avoiding unnecessary antibiotic use may be beneficial for cancer prevention; however, further investigation is needed.

## Pending issues

1. Future research should focus on longitudinal studies with multiple sampling and assessing dietary pattern in breast cancer patients to monitor microbiome changes during treatment.
2. Understanding differences in functional outputs, like microbial genes pathway enrichment and metabolite profiles, could provide critical insights into microbial activities and their impact on clinical outcomes in breast cancer patients across multiple studies.
3. Mechanistic studies are needed to model how gut bacteria influence breast cancer and treatment responses.

## Peer review information

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

## Acknowledgements

Authors of this work are funded by the Department of Defense Breast Cancer Research Program (BC210715 (W81XWH-22-1-0055) to KLC and BC230701 (HT9425-24-1-0025) to KLC and MHM), the National Cancer Institute Grants (R01CA253329 and U01CA272541) to KLC, the V Foundation (D2024-014) to KLC and AH, and the 2024 AACR-AstraZeneca Breast Cancer Fellowship for Endocrine Therapy Research (24-40-12-ARNO) to AAA.

## Author contributions

**Alana A Arnone**: Writing—original draft; Writing—review and editing. **Katherine Ansley**: Writing—original draft. **Arielle L Heeke**: Writing—original draft. **Marissa Howard-McNatt**: Writing—original draft. **Katherine L Cook**: Funding acquisition; Writing—original draft; Writing—review and editing.

## Disclosure and competing interests statement

The authors declare no competing interests.

