## [Peer Review File · EMBO Molecular Medicine]

Gut Microbiota Interact with Breast Cancer Therapeutics to Modulate Efficacy

Alana Arnone, Katherine Ansley, Arielle Heeke, Marissa Howard-McNatt, and Katherine Cook

Corresponding author: Katherine Cook (klcook@wakehealth.edu)

Review Timeline:

Submission Date:	5th Sep 24
Editorial Decision:	7th Oct 24
Revision Received:	26th Nov 24
Editorial Decision:	29th Nov 24
Revision Received:	2nd Dec 24
Accepted:	3rd Dec 24

Editor: Lise Roth

Transaction Report:

7th Oct 2024

Dear Katherine,

Thank you for the submission of your review to EMBO Molecular Medicine, and please accept my apologies for the delay in getting back to you as one referee needed more time to complete his/her review. We have now received feedback from the experts who agreed to evaluate your manuscript.

As you will see from the reports below, they found the review new and interesting, with a good overview of the literature, and are supportive of publication pending minor revisions.

We will therefore welcome the submission of a revised manuscript addressing the referees' comments and suggestions at your earliest convenience.

- 1) a .doc formatted version of the manuscript text (including Figure legends and tables)
- 2) Separate figure files
- 3) a letter INCLUDING the reviewer's reports and your detailed responses to their comments.

Please also include:

- A glossary: EMBO Molecular Medicine articles are accompanied by a glossary explaining some of the terms used for laymen. Glossary is not a list of abbreviations, abbreviations should be defined the first time they are used in the text
- A section on 'Pending issues': At the end of each article, there is a box highlighting issues that still need further studies and where research efforts should converge (called the Pending issues box). This should be limited to a few points.

Regarding the figures, please note the following points:

- If there are certain aspects of your figures that are based upon assumptions or where the scientific data remains ambiguous, please add a comment so that we can work with you on an accurate depiction. Please ensure the directionality and nature of interactions is presented accurately.
- If the figure or single panels of the figure have been adapted from a published figure, please add this information to the figure legend (e.g., 'Adapted from...' or 'Based on...').
- Please only re-use figures or parts of a figure if this is essential for understanding the concept communicated. If the figure contains re-used images or elements of images, please make sure that you have the permission/license to publish it (this also applies to your own previous work, if the journal you published in retains copyright. Certain 'creative commons' open access licenses, such as CC-BY 4.0, allow re-use without additional formal permissions). All re-used material must be explicitly cited.
- If you use an image data base for scientific iconography (e.g., BioRender), please let us know if you have a license that allows for publication in an academic journal. Often authors use misleading iconography for expedience. Please ensure the information shown is scientifically accurate.

Looking forward to receiving your revised manuscript,

With kind regards,

Lise

***** Reviewer's comments *****

Referee #1 (Remarks for Author):

The review touches on a broad range of breast cancer therapies, including specific drugs and their interactions with the gut microbiome, as well as potential impacts on treatment efficacy and toxicity. Citing both preclinical and clinical studies adds depth and highlights the complexity of these interactions. However, several structural and content-related issues need to be improved in order to optimize the relevance and clarity of this timely review.

1. Important- the review should explicitly consider Controversies: Including a brief discussion on challenges or controversies (e.g., individual variability in microbiomes, conflicting probiotic results) would add depth and balance to the review.
2. The review would benefit from clearer subheadings to separate different classes of chemotherapy (e.g., anthracyclines, platinum-based drugs) and their interactions with the microbiome.
3. There is a noticeable gap in transitioning between sections. For example, the shift from preclinical to clinical data in the chemotherapy section is abrupt, and smoother transitions would enhance readability. Introducing sections with transition sentences could create a more cohesive narrative.
4. In multiple places (e.g., pages 8, 9), there are meaningless strings of characters ("W S H P D F U S K J H V "). These need to be corrected for clarity and professionalism.
5. Some references mentioned in the text are missing from the bibliography (e.g., Miko et al., 2019). The bibliography also needs to be better organized, as some references (e.g., Bobin-Dubigeon et al.) are misclassified, such as incorrectly labeled as preprints.
6. Several of the sections lack depth. For instance, when discussing microbial dysbiosis and its link to breast cancer risk ("Microbial dysbiosis increases breast cancer risk, as certain gut bacteria alter the production of beneficial anticancer metabolites and disrupt estrogen metabolism in the gut"), the review does not provide concrete examples. Adding specific bacterial species or mechanisms would give more weight to these discussions.
7. In multiple instances, key claims are not supported by references. For example, the statement about alpha diversity being lower after chemotherapy in a cohort of postmenopausal women (Dutch study) lacks citation, as does the finding about elevated GUS levels in nipple aspirate fluid in breast cancer patients.
8. The review often touches on important points without fully explaining them. For example, the "CANTO prospective study" finding that chemotherapy favors colonization by beneficial commensals needs more explanation on the implications of these microbiome shifts. Similarly, the discussion of probiotics improving doxorubicin efficacy lacks detail, especially since the cited article (Ye et al., 2024) shows only a mild effect.
9. In the section on endocrine therapies, the text jumps between topics like microbial diversity, estrogen metabolism, drug detoxification, and short-chain fatty acids (SCFAs) without clear transitions. This could be resolved by adding linking sentences to improve flow and by organizing the information more logically.
10. In certain sections, the review switches between TNBC (triple-negative breast cancer) and HR+ (hormone receptor-positive) breast cancer without clear delineation. Defining the treatment context and maintaining consistency would help improve focus.
11. The chemotherapy section could benefit from more detailed explanations of how microbiome changes influence toxicity and side effects. This would make the clinical relevance of these findings clearer.
12. Clarify Probiotic and Immunotherapy Discussion: The probiotic section in the immunotherapy discussion needs to be a bit more thoroughly explained, especially how specific probiotic strains may interact with immune-based therapies.
13. Address Missing Mechanistic Details in Obesity Section: The obesity-induced dysbiosis section lacks specific details on how gut dysbiosis impacts immune responses, the tumor microenvironment, and drug metabolism. Further elaboration is needed, especially regarding how dysbiosis affects HER2-targeted therapies and doxorubicin resistance.
14. The review should more explicitly address the potential clinical relevance of microbiome-based interventions, such as fecal microbiota transplantation (FMT) or GUS inhibitors. Adding a conclusion that highlights how microbiome manipulation could improve therapy efficacy and reduce side effects would make the review more impactful.
15. It would be important to highlight how findings from animal models might translate to human applications or outline limitations when necessary. This would add credibility to the review's relevance in clinical practice.
16. The order of topics in the figure does not match the text (e.g., chemotherapy is discussed first in the text but appears second

in the figure). Keeping a consistent order between text and visuals will improve coherence.

17. The figure's resolution is poor and needs to be improved for clearer representation.

Referee #2 (Remarks for Author):

This review focuses on the role of gut microbiota in influencing the pharmacokinetics of chemo- and endocrine therapies and their consequences on anti-tumor efficacy. Additionally, the authors provide an overview of the impact of gut microbiota on other drugs used in breast cancer (BC) treatment, as well as its influence on immune system regulation and obesity.

Overall, the review covers all the major points that have emerged from the interaction between BC and gut microbiota, and it is generally well written. However, some minor revisions and corrections are required to improve the manuscript before acceptance.

1-The authors provide a concise overview of the microbiota's ability to regulate systemic inflammation and immune responses to anti-cancer treatment. Given that this is one of the key points in the gut microbiota-BC interaction, as highlighted by the authors, I recommend including a dedicated section on this topic in the manuscript, rather than limiting it to a general description in the introduction.

2-Page 6: "DOX treatment increased *A. muciniphila* in a murine 4T1 TNBC model, with an associated reduction in tumor weight and metastatic burden." Did DOX increase *A. muciniphila* regardless of treatment efficacy, as its abundance also increased in non-responder mice? I suggest rephrasing this paragraph and adding the information that "by stratifying mice into DOX responders and non-responders, the responders showed elevated *Akkermansia muciniphila* abundance prior to DOX treatment," as reported by Bawaneh et al. (2022). Please make the same adjustment on page 8 when citing Bawaneh et al. (2022).

3-How do the authors address the contrast between the findings of Bawaneh et al. (2022), which showed an increase in *Akkermansia* in preclinical models, and the findings by Bilendike et al. (2022), where breast cancer patients undergoing chemotherapy experienced a decrease in mucin-degrading bacteria like *Akkermansia* compared to healthy controls?

4-Page 7: Please provide more specific details in the statement, "Another study found that the gut microbiota of human epidermal growth factor receptor-2 (HER2)-negative metastatic breast cancer patients receiving metronomic chemotherapy (n=15), referring to chemotherapy administered in low (1/10-1/3 of the maximum tolerated dose), minimally toxic doses on a frequent schedule, was different in terms of diversity, composition, and function from those under conventional chemotherapy (n=16) (Guan et al., 2020)." How were diversity, composition, and function different?

5-Page 12: Please rephrase the end of the first and last paragraphs, as they are repetitive. First: "Further study is needed to determine whether oral endocrine-targeting therapies alter and/or are metabolized by gut microbiota and how that may affect cancer recurrence." Last: "Further study is needed to determine whether oral endocrine therapies used to treat ER+ breast cancer, including tamoxifen, alter and/or are metabolized by gut microbiota and how that may impact cancer recurrence."

6-Please check all citations throughout the manuscript. Several references need to be corrected or replaced with more appropriate citations, as detailed below:

Chemotherapy section

Page 7: A reference appears to be missing in line 3.

Please provide a citation for the CANTO prospective study.

Page 9: Please provide a reference for the role of SCFA butyrate in enhancing IL-12 receptor expression on CD8+ T cells.

Endocrine therapy section

Pages 9-10: Please provide more specific references regarding the clinical practice of ER+ BC treatment, instead of relying on Chen (2011) and Terrisse (2013).

Immunotherapy section

Page 15: "Bacteria metabolites that...can enhance therapeutic efficacy" - Di Modica et al. (2021) is not the appropriate citation here. The authors likely meant to refer to doi: 10.3389/fonc.2022.947188 (PMID: 35912227).

Page 15: Please insert a reference for the impact of TMAO on anti-tumor immunity.

Other targeted therapies section

Page 15: Provide a more suitable reference for the clinical management of HER2+ BC, as Alpuim Costa et al. is more general to microbiome and breast cancer rather than specific to HER2+ BC treatment.

Page 16: "Administration of *Lactococcus lactis* or *Lactobacillus paracasei* with trastuzumab improved its efficacy in mice under vancomycin." These data are not reported in Di Modica et al., Cancer Research 2021, but rather in "Di Modica, Martina (2020).

Obesity-induced gut dysbiosis section

Page 18: "FMT from trastuzumab-responding patients...can enhance or reduce the efficacy of HER2-targeted therapies" - the reference to de Oliveira Andrade is not appropriate. Please provide the correct citation.

Referee #3 (Remarks for Author):

Arnone et al, "Gut Microbiota Interact with Breast Cancer Therapeutics to Modulate Efficacy". This is a very well written comprehensive review which cites appropriate literature, nicely synthesizes the data and puts it into context. Data from human cohorts and from mechanistic mouse model experiments is analyzed. Endocrine-targeting Therapies is a nice and rather unique section for microbiome & cancer review in a context of breast cancer. Overall this review is very close to be in "accept as is" form, only a few suggestions are listed below:

- 1) Have at least some discussion about "Specie, specie and another distinct specie..." vs "common function by different species of bacteria"- what happens more often. The former is probably great for established microbiome therapies "magic 1 specie yogurt and 1 particular metabolite". The latter may also lead to a particular metabolite but also may be relevant for common biological mechanisms.
- 2) Very similarly "Metabolites vs Species" may deserve a separate section?

We would like to thank the reviewers for their comments. We have made the requested revisions to the manuscripts based upon the reviewer and editorial office suggestions. Please see below for point-by-point response to the reviewer's comments.

Referee #1 (Remarks for Author):

The review touches on a broad range of breast cancer therapies, including specific drugs and their interactions with the gut microbiome, as well as potential impacts on treatment efficacy and toxicity. Citing both preclinical and clinical studies adds depth and highlights the complexity of these interactions. However, several structural and content-related issues need to be improved in order to optimize the relevance and clarity of this timely review.

1. Important- the review should explicitly consider Controversies: Including a brief discussion on challenges or controversies (e.g., individual variability in microbiomes, conflicting probiotic results) would add depth and balance to the review.

Response: We appreciate the Reviewer's input and agree that this would be beneficial to the manuscript. We have added a section discussing current challenges regarding the impact of breast cancer therapies on the gut microbiome and vice versa. We were constrained by overall word limitations however think that this suggestion is critical.

2. The review would benefit from clearer subheadings to separate different classes of chemotherapy (e.g., anthracyclines, platinum-based drugs) and their interactions with the microbiome.

Response: We agree with the reviewer and have added subheadings to better separate the different classes of chemotherapy.

3. There is a noticeable gap in transitioning between sections. For example, the shift from preclinical to clinical data in the chemotherapy section is abrupt, and smoother transitions would enhance readability. Introducing sections with transition sentences could create a more cohesive narrative.

Response: Thank you for highlighting this. We agree that smoother transitions would enhance the flow and readability, especially in bridging preclinical and clinical findings. We have incorporated transition sentences to guide readers through these shifts more seamlessly, ensuring each section connects logically to the next.

4. In multiple places (e.g., pages 8, 9), there are meaningless strings of characters ("W□S□H□P□D□F□U□S□K□J□H□V□□"). These need to be corrected for clarity and professionalism.

Response: Thank you for bringing this to our attention. We did not have these characters in the original text, which makes us think that there was an issue in the upload/conversion. We will double check this current uploaded version to confirm that this issue was not repeated.

5. Some references mentioned in the text are missing from the bibliography (e.g., Miko et al., 2019). The bibliography also needs to be better organized, as some references (e.g., Bobin-Dubigeon et al.) are misclassified, such as incorrectly labeled as preprints.

Response: Thank you for highlighting this. We have fixed any discrepancies with the references.

6. Several of the sections lack depth. For instance, when discussing microbial dysbiosis and its link to breast cancer risk ("Microbial dysbiosis increases breast cancer risk, as certain gut bacteria alter the production of beneficial anticancer metabolites and disrupt estrogen metabolism in the gut"), the review does not provide concrete examples. Adding specific bacterial species or mechanisms would give more weight to these discussions.

Response: Thank you for this valuable feedback. We agree that adding specific examples of bacterial species and mechanisms would provide greater depth and clarity. We were constrained by overall word limitations however think that this suggestion is critical. In response, we have provided more detailed examples, throughout the manuscript. See below for the details we have added regarding the link between microbial dysbiosis and breast cancer risk.

"Dysbiosis, or microbial imbalance, can elevate breast cancer risk by disrupting estrogen metabolism and reducing the production of anticancer metabolites (Mikó, Kovács et al. 2019, Ruo, Alkayyali et al. 2021). This dysbiosis can lead to chronic inflammation, DNA damage, and activation of oncogenic pathways, allowing bacteria like *Fusobacterium nucleatum* and certain strains of *Escherichia coli* to promote carcinogenesis (Doocey, Finn et al. 2022, Li 2023). Furthermore, intratumoral bacteria can affect cancer phenotypes, such as enhancing the metastatic ability of malignant cells, thereby contributing to disease advancement (Zhao, Mei et al. 2023). Dysregulated bacteria produce metabolites like lipopolysaccharide (LPS) and colibactin, which drive cancer progression through inflammatory and mutagenic pathways. LPS, produced by Gram-negative bacteria like *Escherichia coli*, promotes tumor growth (Soto-Pantoja, Gaber et al. 2021), while colibactin from *E. coli* can cause DNA double-strand breaks, leading to mutations (Sun, Chen et al. 2023). Dysbiosis also depletes beneficial bacteria that produce short-chain fatty acids (SCFAs), such as *Faecalibacterium prausnitzii* and *Roseburia* species, thereby reducing SCFAs' anti-inflammatory and anticancer effects (Shrode, Knobbe et al. 2023).

The depletion of beneficial bacteria, such as SCFA-producing species(Álvarez-Mercado, Del Valle Cano et al. 2023), not only weakens the body's natural defense mechanisms by compromising immune system homeostasis but also creates an environment where pathogenic bacteria may have a greater influence on cancer progression(Doocey, Finn et al. 2022, Lu and Tong 2024).”

7. In multiple instances, key claims are not supported by references. For example, the statement about alpha diversity being lower after chemotherapy in a cohort of postmenopausal women (Dutch study) lacks citation, as does the finding about elevated GUS levels in nipple aspirate fluid in breast cancer patients.

Response: Thank you for pointing this out. We have reviewed the manuscript to ensure all claims are properly cited with relevant studies. We have provided the specific references for these examples, including the Dutch cohort study on chemotherapy's effects on gut microbiota diversity and research on GUS levels in breast cancer patients. We appreciate your attention to this detail.

8. The review often touches on important points without fully explaining them. For example, the "CANTO prospective study" finding that chemotherapy favors colonization by beneficial commensals needs more explanation on the implications of these microbiome shifts. Similarly, the discussion of probiotics improving doxorubicin efficacy lacks detail, especially since the cited article (Ye et al., 2024) shows only a mild effect.

Response: Thank you for your insightful comments. We appreciate your feedback and have revised the manuscript to provide more detailed explanations of the points you've highlighted. Specifically, we have expanded on the implications of the microbiome shifts observed in the CANTO prospective study, elaborating on how chemotherapy's promotion of beneficial commensals might influence treatment outcomes and potentially improve prognosis. Additionally, we have clarified the discussion of probiotics enhancing doxorubicin efficacy, addressing the mild effect reported in Ye et al. (2024) and providing context for the potential, albeit limited, impact of probiotics in chemotherapy treatment.

9. In the section on endocrine therapies, the text jumps between topics like microbial diversity, estrogen metabolism, drug detoxification, and short-chain fatty acids (SCFAs) without clear transitions. This could be resolved by adding linking sentences to improve flow and by organizing the information more logically.

Response: Thank you for your feedback. We agree that clearer transitions and a more organized structure could improve the flow and coherence of this section.

10. In certain sections, the review switches between TNBC (triple-negative breast cancer) and HR+ (hormone receptor-positive) breast cancer without clear delineation. Defining the treatment context and maintaining consistency would help improve focus.

Response: Thank you for your insightful comment. We have revised the manuscript to more clearly delineate between the two subtypes, ensuring that the treatment context for each is clearly defined when switching between them.

11. The chemotherapy section could benefit from more detailed explanations of how microbiome changes influence toxicity and side effects. This would make the clinical relevance of these findings clearer.

Response: Thank you for your suggestion. We agree that providing more detailed explanations of how microbiome changes influence toxicity and side effects in the chemotherapy section would enhance the clinical relevance of these findings. We have revised this section to include additional details on how specific microbial shifts contribute to treatment-related adverse effects.

12. Clarify Probiotic and Immunotherapy Discussion: The probiotic section in the immunotherapy discussion needs to be a bit more thoroughly explained, especially how specific probiotic strains may interact with immune-based therapies.

Response: Thank you for your valuable feedback. We agree that further clarification is needed regarding the interaction between specific probiotic strains and immune-based therapies. In the revised manuscript, we have expanded on how certain probiotic strains, such as *Lactobacillus* and *Bifidobacterium* species, may enhance immune checkpoint blockade (ICB) therapies by modulating the gut microbiome and activating key immune cells like NK cells and T cells. We also discussed how these probiotics can influence immune responses, including promoting Th1-biased immune responses and improving dendritic cell activation, which could potentially enhance the efficacy of immunotherapies like PD-L1 inhibitors. This additional explanation will provide a clearer understanding of the mechanisms through which probiotics may interact with and complement immunotherapy. We have also added the current controversy in the field regarding the disparate outcomes in some studies that investigated probiotic usage and immunotherapy responsiveness in clinical cohorts.

13. Address Missing Mechanistic Details in Obesity Section: The obesity-induced dysbiosis section lacks specific details on how gut dysbiosis impacts immune responses, the tumor microenvironment, and drug metabolism. Further elaboration is needed, especially regarding how dysbiosis affects HER2-targeted therapies and doxorubicin resistance.

Response: Thank you for the feedback. We agree that additional mechanistic details would strengthen the discussion on obesity-induced dysbiosis. We have rewritten this section to provide further insights into how obesity-specific dysbiosis can alter immune responses, the tumor microenvironment, and drug metabolism to influence breast

cancer outcomes. However, we are constrained by overall word limitations so further details regarding obesity and breast cancer risk are expanded upon in a different review (Gaber et. al., 2024).

14. The review should more explicitly address the potential clinical relevance of microbiome-based interventions, such as fecal microbiota transplantation (FMT) or GUS inhibitors. Adding a conclusion that highlights how microbiome manipulation could improve therapy efficacy and reduce side effects would make the review more impactful.

Response: Thank you for the suggestion to expand on the clinical relevance of microbiome-based interventions. We have added a section to emphasize how strategies like fecal microbiota transplantation (FMT) and probiotics hold the potential to improve cancer therapy efficacy and mitigate side effects.

“Microbiome interventions, including fecal microbiota transplants (FMT), probiotics, and dietary modifications, have shown potential to boost immune checkpoint inhibitor response rates by promoting favorable microbial compositions. These approaches may also reduce immune-related side effects, such as colitis, by modulating inflammatory responses. Additionally, microbiome-based strategies could help overcome therapy resistance, with studies indicating FMT and probiotics may re-sensitize resistant tumors. Personalized microbiome profiling could further tailor these interventions, maximizing benefits while enhancing chemotherapy’s efficacy and reducing its toxicity.”

15. It would be important to highlight how findings from animal models might translate to human applications or outline limitations when necessary. This would add credibility to the review's relevance in clinical practice.

Response: Thank you for the comment. We have added a section discussing the potential limitations associated with using animal models. See below.

Current Limitations:

While animal models provide valuable insights, the human gut microbiome is considerably more complex and diverse, shaped by unique diets, environments, and physiological factors that may limit the direct applicability of animal findings(Sundberg and Schofield 2018). Studies show that breast cancer patients often have altered gut microbiota, though the specific changes and impacts vary widely among individuals, making it challenging to develop universal microbiome-based interventions(Nandi, Parida et al. 2023). Many animal studies focus on short-term outcomes, while breast cancer treatments in humans typically last months or years, leaving the long-term effects of microbiome modulation on treatment efficacy

and side effects largely unknown. Although links between gut microbiota and treatment outcomes have been observed, the exact mechanisms by which gut bacteria influence breast cancer progression and response to treatment in humans are not yet fully understood. To bridge this gap, future research should prioritize larger, longitudinal studies in human breast cancer patients to monitor microbiome changes during treatment, develop personalized microbiome interventions based on patient profiles, investigate the safety and efficacy of microbiome-modulating therapies in clinical trials, and clarify the specific mechanisms by which gut bacteria impact breast cancer and treatment response. Expanding our knowledge of the gut microbiome's role in breast cancer could pave the way for more effective, microbiome-informed therapies for patients.

16. The order of topics in the figure does not match the text (e.g., chemotherapy is discussed first in the text but appears second in the figure). Keeping a consistent order between text and visuals will improve coherence. The figure's resolution is poor and needs to be improved for clearer representation.

Response: Thank you for your valuable feedback. We have made the necessary adjustments to ensure the order of topics in the figure matches the sequence presented in the text, starting with chemotherapy. Additionally, we have improved the resolution of the figure to ensure clearer representation and better readability. We appreciate your attention to these details.

Referee #2 (Remarks for Author):

This review focuses on the role of gut microbiota in influencing the pharmacokinetics of chemo- and endocrine therapies and their consequences on anti-tumor efficacy. Additionally, the authors provide an overview of the impact of gut microbiota on other drugs used in breast cancer (BC) treatment, as well as its influence on immune system regulation and obesity.

Overall, the review covers all the major points that have emerged from the interaction between BC and gut microbiota, and it is generally well written. However, some minor revisions and corrections are required to improve the manuscript before acceptance.

1-The authors provide a concise overview of the microbiota's ability to regulate systemic inflammation and immune responses to anti-cancer treatment. Given that this is one of

the key points in the gut microbiota-BC interaction, as highlighted by the authors, I recommend including a dedicated section on this topic in the manuscript, rather than limiting it to a general description in the introduction.

Response: Thank you for your comment. We were constrained by overall word limitations of the journal, and therefore the potential interactions between the gut microbiome and immune cell programming are discussed in the context of each therapy individually based upon availability of published literature and not in its own section. We found this helps the flow of the manuscript and reduced the word count.

2-Page 6: "DOX treatment increased *A. muciniphila* in a murine 4T1 TNBC model, with an associated reduction in tumor weight and metastatic burden." Did DOX increase *A. muciniphila* regardless of treatment efficacy, as its abundance also increased in non-responder mice? I suggest rephrasing this paragraph and adding the information that "by stratifying mice into DOX responders and non-responders, the responders showed elevated *Akkermansia muciniphila* abundance prior to DOX treatment," as reported by Bawaneh et al. (2022). Please make the same adjustment on page 8 when citing Bawaneh et al. (2022).

Response: Thank you for your suggestion. We agree that clarifying the relationship between *A. muciniphila* abundance and treatment efficacy is important for better understanding the findings. We have rephrased the paragraph to reflect that *A. muciniphila* abundance was elevated prior to DOX treatment in the responder group.

3-How do the authors address the contrast between the findings of Bawaneh et al. (2022), which showed an increase in *Akkermansia* in preclinical models, and the findings by Bilendike et al. (2022), where breast cancer patients undergoing chemotherapy experienced a decrease in mucin-degrading bacteria like *Akkermansia* compared to healthy controls?

Response: Thank you for the comment. We have addressed differences in findings between Bawaneh et al. (2022) and Bilendike et al. (2022) outlined below.

"These findings stand in contrast to those observed in breast cancer patients undergoing

chemotherapy, who experienced a decrease in *Akkermansia*. Such discrepancies may arise

from species-specific responses between mice and humans, as well as from treatment

differences: Bawaneh et al. focused on the effects of doxorubicin (DOX) alone, whereas

Bilendike et al. examined seven different chemotherapy combinations and sequences.

Additional factors also play a role, including baseline microbiome differences, as breast cancer

patients generally have distinct microbiomes compared to healthy individuals or preclinical

models(Nandi, Parida et al. 2023). Host factors like diet, lifestyle, and genetic background

further contribute to variability, as human patients have more diverse profiles than laboratory mice, which can influence how the microbiome responds to cancer treatments (Org and Lusic 2018). Finally, differences in sample collection and analysis techniques could also account for these variations (Liu, Zhong et al. 2024)."

4-Page 7: Please provide more specific details in the statement, "Another study found that the gut microbiota of human epidermal growth factor receptor-2 (HER2)-negative metastatic breast cancer patients receiving metronomic chemotherapy (n=15), referring to chemotherapy administered in low (1/10-1/3 of the maximum tolerated dose), minimally toxic doses on a frequent schedule, was different in terms of diversity, composition, and function from those under conventional chemotherapy (n=16) (Guan et al., 2020)." How were diversity, composition, and function different?

Response: Thank you for the comment. We have expanded on the observed differences in diversity, composition, and function in the gut microbiota between patients on metronomic chemotherapy and those on conventional chemotherapy, as reported by Guan et al., 2020. Specifically, we described how diversity differed, highlighting reduced microbial diversity in the metronomic chemotherapy cohort. Additionally, we have detailed specific taxa that varied in abundance, such as *Blautia obeum* and *Slackia*, and explained their functional roles and associations with progression-free survival (PFS). This added detail will provide a clearer understanding of how metronomic dosing impacts the gut microbiome compared to conventional chemotherapy. See below.

"A study examining the gut microbiota of HER2-negative metastatic breast cancer patients found distinct differences in composition and function between those receiving metronomic chemotherapy with Capecitabine (n=15) — a low, minimally toxic dose given frequently — and those on a conventional chemotherapy regimen (n=16). The metronomic group showed reduced gut microbiota diversity compared to the conventional group. At the phylum level, patients in both dosage groups showed similar microbial compositions, with Bacteroidetes, Firmicutes, Proteobacteria, and Actinobacteria making up over 95% of the microbiome. At the genus level, the top five genera were Bacteroides, Prevotella, Roseburia, Faecalibacterium, and *f_Lachnospiraceae*, contributing 32.9%, 12.2%, 6.9%, 6.2%, and 5.6% of the microbiome in the metronomic group and 35.3%, 12.3%, 7.3%, 6.8%, and 6.8% in the routine group, respectively. These genera were slightly more abundant in the routine group.

Additionally, *Megamonas* and *f_Mogibacteriaceae* were enriched in the metronomic group, while *Blautia* and *o_Streptophyta* were reduced. Notably, in patients receiving metronomic chemotherapy, higher abundance of *Blautia obeum* was associated with significantly longer progression-free survival (PFS; 32.7 vs. 12.9 months, $P = 0.013$), whereas the presence of *Slackia* correlated with shorter PFS (9.2 vs. 32.7 months, $P = 0.004$). *B. obeum* is implicated in the transformation of carcinogenic heterocyclic amines, with decreased levels potentially increasing cancer risk. Conversely, *Slackia* has been linked to colorectal cancer development in several studies, indicating its potential as a microbial biomarker for cancer prevention, diagnosis, and treatment (Guan, Ma et al. 2020)."

5-Page 12: Please rephrase the end of the first and last paragraphs, as they are repetitive. First: "Further study is needed to determine whether oral endocrine-targeting therapies alter and/or are metabolized by gut microbiota and how that may affect cancer recurrence." Last: "Further study is needed to determine whether oral endocrine therapies used to treat ER+ breast cancer, including tamoxifen, alter and/or are metabolized by gut microbiota and how that may impact cancer recurrence."

Response: Thank you for pointing out the repetition. We have rephrased these sentences to make them distinct while preserving the meaning. We have revised the first instance to focus on the potential impact of gut microbiota on therapy efficacy and the second to emphasize future research directions on the interplay between endocrine therapies and gut microbiota in cancer recurrence. See below:

First: "However, further research is needed to understand how gut microbiota may metabolize or alter oral endocrine-targeting therapies, potentially influencing their efficacy and affecting cancer recurrence."

Last: "Future research should investigate the interplay between oral endocrine therapies, such as tamoxifen, and the gut microbiota in ER+ breast cancer, exploring how microbial metabolism of these therapies may impact their efficacy and influence cancer recurrence."

6-Please check all citations throughout the manuscript. Several references need to be corrected or replaced with more appropriate citations, as detailed below:

Chemotherapy section

Page 7: A reference appears to be missing in line 3.

Please provide a citation for the CANTO prospective study.

Page 9: Please provide a reference for the role of SCFA butyrate in enhancing IL-12 receptor expression on CD8+ T cells.

Endocrine therapy section

Pages 9-10: Please provide more specific references regarding the clinical practice of ER+ BC treatment, instead of relying on Chen (2011) and Terrisse (2013).

Immunotherapy section

Page 15: "Bacteria metabolites that...can enhance therapeutic efficacy" - Di Modica et al. (2021) is not the appropriate citation here. The authors likely meant to refer to doi: 10.3389/fonc.2022.947188 (PMID: 35912227).

Page 15: Please insert a reference for the impact of TMAO on anti-tumor immunity.

Other targeted therapies section

Page 15: Provide a more suitable reference for the clinical management of HER2+ BC, as Alpuim Costa et al. is more general to microbiome and breast cancer rather than specific to HER2+ BC treatment.

Page 16: "Administration of Lactococcus lactis or Lactobacillus paracasei with trastuzumab improved its efficacy in mice under vancomycin." These data are not reported in Di Modica et al., Cancer Research 2021, but rather in "Di Modica, Martina (2020). Gut Microbiota and Trastuzumab Response in HER2-Positive Breast Cancer. PhD thesis, The Open University."

Obesity-induced gut dysbiosis section

Page 18: "FMT from trastuzumab-responding patients...can enhance or reduce the efficacy of HER2-targeted therapies" - the reference to de Oliveira Andrade is not appropriate. Please provide the correct citation.

Response: Thank you for bringing this to our attention. We have carefully reviewed each citation to ensure it is accurate and aligns with the content it supports. We have also replaced any references with more relevant sources where necessary, based on the specific details provided.

Referee #3 (Remarks for Author):

Arnone et al, "Gut Microbiota Interact with Breast Cancer Therapeutics to Modulate Efficacy". This is a very well written comprehensive review which cites appropriate literature, nicely synthesizes the data and puts it into context. Data from human cohorts and from mechanistic mouse model experiments is analyzed. Endocrine-targeting Therapies is a nice and rather unique section for microbiome & cancer review in a context of breast cancer. Overall this review is very close to be in "accept as is" form, only a few suggestions are listed below:

1) Have at least some discussion about "Specie, specie and another distinct specie..." vs "common function by different species of bacteria"- what happens more often. The former is probably great for established microbiome therapies "magic 1 specie yogurt and 1 particular metabolite". The latter may also lead to a particular metabolite but also

may be relevant for common biological mechanisms.

2) Very similarly "Metabolites vs Species" may deserve a separate section?

Response: Thank you for your review and comments. We have expanded upon the limitations section and clinical perspective section. It is important to note that there is high variability in clinical gut microbiome cohorts and that species abundance differences may not be as important as functional outputs (ie. Gut microbiome gene differences in starch metabolism) that may highlight more similarities across studies. This is the same for metabolite generation as multiple different types of microorganisms will produce similar metabolites that may be more representative across studies. We have also included discussion on the importance of incorporating dietary assessment into clinical data on gut microbiome as this will not only influence gut microbiome populations but functional outputs as well.

29th Nov 2024

Dear Katherine,

Thank you for submitting your revised manuscript. I have looked at the files and almost everything is fine now. I will therefore be able to accept your manuscript once the following is addressed:

The figure remains difficult to read. We thought it could be re-arranged for better visibility, i.e. instead of having the 4 treatment types in a line, they could be in a 'square' (2 on top and 2 in the bottom). I would also suggest increasing the font for better legibility.

In the legend, please define each abbreviation and element (cells, bacteria, etc).

Once this is done, please submit your revised files for acceptance.

Looking forward to receiving your revised manuscript,

With my best wishes,

Lise

The authors addressed the remaining formatting issues.

3rd Dec 2024

Dear Katherine,

I am pleased to inform you that your manuscript is accepted for publication and is now being sent to our publisher to be included in the next available issue of EMBO Molecular Medicine!

Your manuscript will be processed for publication by EMBO Press. It will be copy edited and you will receive page proofs prior to publication. Please note that you will be contacted by Springer Nature Author Services to complete licensing information.

There is no charge for this Review Article, but in a few weeks when you are contacted to sign your license agreement and review article proofs, please enter this token into the appropriate field in the Springer Nature author services system: XXXXXXXXXXXXX.

If you have any questions, please do not hesitate to contact the Editorial Office.

Thank you for contributing this interesting and timely review for EMBO Molecular Medicine!

With kind regards,

Lise
